# High-Velocity Impacts of Pyrophoric Alloy Fragments on Thin Armour Steel Plates

**DOI:** 10.3390/ma14164649

**Published:** 2021-08-18

**Authors:** Evaristo Santamaria Ferraro, Marina Seidl, Tom De Vuyst, Norbert Faderl

**Affiliations:** 1French-German Research Institute of Saint Louis (ISL), 68300 Saint Louis, France; marina.seidl@isl.eu (M.S.); norbert.faderl@isl.eu (N.F.); 2School of Physics, Engineering and Computer Science, University of Hertfordshire, Hatfield AL109EU, UK; t.de-vuyst@herts.ac.uk

**Keywords:** reactive materials, impact-induced energy release, high-velocity impact, pyrophoric alloys

## Abstract

The terminal ballistics effects of Intermetallic Reactive Materials (IRM) fragments have been the object of intense research in recent years. IRM fragments flying at velocities up to 2000 m/s represent a realistic threat in modern warfare scenarios as these materials are substituting conventional solutions in defense applications. The IRM add Impact Induced Energy Release (IIER) to the mechanical interaction with a target. Therefore, the necessity of investigations on IIER to quantify potential threats to existing protection systems. In this study, Mixed Rare Earths (MRE) fragments were used due to the mechanical and pyrophoric affinity with IRM, the commercial availability and cost-effectiveness. High-Velocity Impacts (HVI) of MRE were performed at velocities ranging from 800 to 1600 m/s and recorded using a high-speed camera. 70 MREs cylindrical fragments and 24 steel fragments were shot on armour steel plates with thicknesses ranging from 2 mm to 3 mm. The influence of the impact pitch angle (α) on HVI outcomes was assessed, defining a threshold value at α of 20°. The influence of the failure modes of MRE and steel fragments on the critical impact velocities (*CIV*) and critical kinetic energy ($E_{\mathrm{kin}\;\mathrm{crit}}$) was evaluated. An energy-based model was developed and fitted with sufficient accuracy the Normalised $E_{\mathrm{Kin}\;\mathrm{crit}}$ (${\tilde{E}}_{kin}^{crit}$) determined from the experiments. IIER was observed in all the experiments involving MRE. From the analyses, it was observed that the IIER spreads behind the targets with velocities comparable to the residual velocities of plugs and shattered fragment.

## 1. Introduction

The interest of the defence industry towards Intermetallic Reactive Materials (IRM) is related to their structural properties, combined with the exothermal reaction triggered by a thermal or mechanical shock, as discussed by Aydelotte [1].

IRM represent a valid substitute for inert casings of next-generation warheads, augmenting the post-detonation lethality abundantly. The scheme in Figure 1 shows how small IRM fragments are generated and accelerated: after the detonation (a), the IRM bomb casing fractures into fragments (b), which are then accelerated by the detonation gases and the blast wave (c), reaching velocities up to 2000 m/s and covering a lethality range significantly higher than the one covered by the blast, as Aydelotte [1] reported. The IRM fragments will eventually interact with a target, triggering the Impact Induced Energy Release (IIER) (d). 

The IIER of fragments is the focus of this work. In order to set up a coherent and reproducible methodology, High-Velocity Impacts (HVI) experiments were performed using a ballistic set-up, shooting commercial Mixed Rare Earths (MREs) cylindrical samples, with diameters of 3.5 mm and 5 mm and length *L* over diameter *d* ratios (*L/d*) of 1. 

MREs fragments were selected to perform the experiments considering their affinity with IRM commonly described in the relevant literature, such as the nickel-aluminium (Ni-Al) based investigated by Aydelotte et al. [1,2] and Beason et al. [3], the aluminium-tungsten (Al-W) based RMs assessed by Aydelotte et al. [1,2], or the tungsten-zirconium (W-Zr) based intermetallics described by Zhang et al. [4]. Furthermore, both IRM and MREs are brittle, according to the definition provided by Lemaitre et al. [5], and therefore, these materials shatter during high-velocity impacts, reacting exothermically and releasing energy in the form of heat.

The ballistic and pyrophoric properties of commercial MREs with different compound compositions were investigated by Waite et al. [6]. In the study, thirty-three metals and metal alloys were assessed and compared. Cylindrical fragments were shot with velocities ranging from 300 m/s to 1600 m/s. The targets were 1.27 mm thick 7075-T6 aluminium plates (HB 150), 1.6 mm thick titanium plates (HB 145) and 1.8 mm thick 1010 steel (HB 100). All the MREs mixtures assessed showed IIER, recording temperatures ranging from 2300 °C up to 3000 °C [6].

Similar observations were described by Hillstrom [7], who assessed the ignition threshold for cylindrical fragments of MREs impacting 6.35 mm thick aluminium targets (HB 120) and 38 mm thick steel blocks (HB 140). Both Waite et al. [6] and Hillstrom [7] documented that commercial MREs showed mechanically induced pyrophoricity at lower impact velocities than other pyrophoric metals. The authors linked the IIER with the failure mechanisms observed: the pyrophoric metals tested, such as zirconium, titanium, hafnium, steel or copper, experience ductile failure and mechanical induced ignition from the frictional stresses caused by the target/penetrator interaction. On the other hand, the ignition mechanism for the MREs samples was attributed to internal shear stresses producing intergranular friction and heating. 

Furthermore, Hillstrom [7] noted that the thickness and material of the target influenced the IIER for MREs. The different mechanical properties involved in the fragment/plate interaction affected the values of impulsive load applied to the fragment, influencing the shattering and reaction.

Aydelotte et al. [1,2] and Beason et al. [3], among others, linked impact-induced fracture of IRM to IIER. In the case of MREs, an oxide reduction is responsible for the IIER, while the reaction experienced by IRM is an intermetallic formation reaction, manifesting as heat release rather than pressure rise, as Aydelotte [1] and Cagle [8] observed. However, even considering the significantly different nature of the reactions, the macroscopic effects of IIER are considerably similar, and peak temperatures are in the same range. Therefore, MREs fragments represent a valid selection for the investigation of IIER during HVIs. In addition, the commercial availability and relative cost-effectiveness make MREs a sensible choice for the study. 

The works from Waite et al. [6] and Hillstrom [7] represent a valid reference in assessing the ballistic and pyrophoric investigation of MREs. The present work discusses the IIER of MREs using a state-of-the-art experimental set-up, described in detail in Section 2.1. Furthermore, the HVI of MREs described in this paper were performed using armour steel plates with HB hardness significantly higher than targets used in previous studies, as they are representative of a realistic target in a modern warfare scenario. 

The experimental outcomes were quantified in terms of Critical Impact Velocities (*CIV*), Critical Kinetic Energy ($E_{\mathrm{kin}\;\mathrm{crit}}$), Specific Kinetic Energy ($E_{Kin\;crit}^{Specific}$) and normalised $E_{\mathrm{Kin}\;\mathrm{crit}}$ (${\tilde{E}}_{kin}^{crit}$). Prior to the quantitative analysis of the HVI outcomes, the influence of the impact pitch angle (α) on the perforation process was investigated to define a threshold value for α. Details are discussed in Section 2.2. 

The development of an energy-based model is exposed in Section 2.3 and Section 4.

The investigation of the IIER is discussed in Section 3.4.

## 2. Materials and Methods

HVI experiments of MRE fragments impacting armour steel plates were performed using a 7.62 mm calibre powder gun. The set-up allows the fragments to reach velocities up to 1600 m/s. A double infrared (IR) light barrier LS 260 was employed as impact velocity measurement system for the fragments before the impact. The IR were 0.5 m apart and can capture velocities up to 2000 m/s. A Shimadzu HPV-1 high-speed camera, triggered by the IR system, records the shots. The target is installed inside a closed ballistic chamber, equipped with a window allowing high-speed recording.

Among the parameters affecting the fragment/plate interaction, the impact pitch angle (α), defined as the angle formed between the longitudinal axis of the fragment and the horizontal flight direction, plays a crucial role, as discussed by Zukas [9] and Rosenberg et al. [10]. The experimental set-up, shown schematically in Figure 2, enables measuring α but does not allow the measurement of the yaw angle, i.e., the angle between the longitudinal axis of the fragment and the vertical reference plane, nor any rolling rotation.

By analyzing the high-speed recordings it was possible measuring the residual velocities ($v_{res}$) reached by the shattered fragments behind the target after perforation, by the plugs ejected from the plate after the perforation, and by the energy release front. The evaluation of $v_{res}$ was performed by measuring the component of the velocity vector normal to the target, as indicated in Figure 3.

The effects of α were quantified by measuring the significant variations in $v_{res}$, using as reference a 10 mm × 10 mm grid placed on the background of the ballistic chamber: the grid as allows estimating the distance travelled in a specific timeframe. The recordings were performed at a frame rate of 250,000 fps.

For reference, Figure 3 shows 3 frames from the high speed recordings of a 5 mm MRE fragment impacting a 2 mm plate at 889 m/s. It is visible that the plug, the front of the debris cloud and the energy release front, travel with significantly similar residual velocities, provided the absence of intense rotations.

Five shots were performed without lighting to focus on the IIER. This set-up does not allow evaluating the angle of impact α but is necessary to focus on reaction initiation and evolution.

### 2.1. Materials

Mixed Rare Earths (MRE) are commercially available pyrophoric mixtures. The compound used in the experiments is composed mainly of cerium (Ce, 49%) and lanthanum (La, 23%). A detailed list of the components of commercially available MRE from different suppliers is available [11].

Table 1 lists the mechanical properties of MRE and armour steel used in this paper.

A total of 94 cylindrical fragments were used to perform the analyses. The fragments can be divided into three categories: MRE samples with a diameter of 5 mm, MRE samples with a diameter of 3.5 mm, and steel fragments with a diameter of 4.6 mm. All fragments had a length (*L*) over diameter (*d*) ratio (*L/d*) of one. The masses of the MRE fragments are 0.6 and 0.2 g, respectively, while the steel fragments weigh 0.6 g. The fragments were encapsulated in a plastic sabot, which in turn was fixed on a 7.62 mm cartridge. The shooting velocities were controlled by adjusting the amount of gun powder used to fill the cartridge. During the flight, the fragment separates from the sabot due to the different kinetic energies.

Similarly to what described by Waite et al. [6] and Hillstrom [7], steel fragments were used in the study. Different solutions were evaluated to achieve high comparability between shots involving fragments made from different materials. In the first approach, the geometrical features and masses were kept constant. In this manner, the experiments had the same initial kinetic energy and the same impact surface. In order to achieve this, considering the different densities of the materials, a hole was drilled on the back end of steel cylinders with diameters of 5 mm and L/d ratio of one to reduce the mass to the desired value. However, this solution was discarded as it influenced the deformation and failure of the fragments significantly. Consequently, an agreement was made to keep the mass of the fragments constant, keeping the *L/d* ratio fixed to one. Therefore, the diameter of the steel fragments was decreased to 4.6 mm steel fragments obtaining a mass of 0.6. Armour steel plates with three different thicknesses (2 mm, 2.5 mm and 3 mm) were used in the experiments. Figure 4 shows a 10 mm × 10 mm × 2 mm plate and the different fragments employed.

### 2.2. Influence of Impact Pitch Angle (α) on HVIs

The impact pitch angle (α) plays a crucial role on impacts, as discussed by Zukas [9] and Rosenberg et al. [10]. However, Zukas [9] highlighted that the influence of α is inhibited when materials with significantly different hardness values interact, as in the case of rigid penetrators impacting a relatively soft target. Similarly, the hardness of the plates used in the experiments described is up to three times (270 ÷ 380 HB) the hardness of the MRE fragments (120 HB). Therefore, the influence of α on the experimental outcomes needed to be quantified. An analysis of the $v_{res}$ of the plugs recorded for 5 mm MRE fragments impacting 2 mm plates was performed to quantify the threshold value of α. 

The Recht-Ipson (RI) [12] formula, shown in Equation (1), was used to fit the experimental data
(1)$$v_{res}=a*{\left(v_{i}^{b}-CIV^{b}\right)}^{\frac{1}{b}}$$
where *a* and *b* are fitting parameters. 

The *CIV* indicates the perforation capabilities of a fragment. *CIV* was defined as the statistically determined minimum velocity necessary to perforate a particular target with a specific fragment, with no residual velocity detected. The definition of perforation is central in this context: for our purpose, a complete perforation is considered when a plug is entirely detached from the plate, and IIER is clearly visible in the form of a reaction spreading behind the target. The *CIV* is evaluated summing the four highest impact velocities for shots that result in no complete perforation to the four lowest impact velocities resulting in complete perforation of the target, dividing the result by the total number of shots considered, as expressed in Equation (2)
(2)$$CIV=\overset{n}{\underset{i=1}{\sum }}\frac{v_{i}^{perf}+v_{i}^{non\;perf}}{2n}$$
where $v_{i}^{perf}$ represents the $i^{th}$ shot resulting in perforation and $v_{i}^{non\;perf}$ represents the $i^{th}$ shot resulting in no perforation of the plate; 2n represents the total number of shots used to evaluate the *CIV*, which in this work is 8.

The Least Square Method was used to perform the curve fitting. It was observed that, for values of α lower than 30° degrees, the effects on residual velocities do not influence the RI curve fit, as visible in Figure 5. It was also observed that higher values of α resulted in lower $v_{res}$, indicating an grater amount of energy dissipated by the target.

In the graph, the blue markers indicate the residual velocities measured for shots impacting the target with 0 degrees, the black markers indicates the residual velocities for impacts at α up to 30° degrees, and the red markers indicate impacts at values of α higher than 30° degrees. A significant effect in the outcomes is evident on the red markers.

The RI curves depicted in Figure 5 were defined by fitting the different dataset represented in the graph. In particular, the blue curve was obtained by fitting the residual velocities from normal impacts, the black curve from all the shots up to 30° degrees, and the red curve by considering the entire data set, including shots having impact angles higher than 30° degrees. The curves are sufficiently similar. However, the residual velocities indicated by the red markers are evidently lower than the value predicted by the RI curves.

Therefore, it is reasonable to use α higher than 30° degrees as exclusion criterion in the analytical analysis of the problem. Bratton et al. [13] described a similar threshold value for IRM impacting 4130 steel targets

### 2.3. Analysis of Critical Impact Energy

The *critical impact velocity (CIV)*, *critical kinetic energy* ($E_{Kin\;crit}$) and residual velocity quantify the ballistic properties of HVIs of the fragments assessed. The $E_{Kin\;crit}$ represents the minimum kinetic energy necessary for the fragment to perforate a specific target. The $E_{Kin\;crit}$ is obtained by imposing the *CIV* value as initial velocity, as described by Equation (3).
(3)$$E_{Kin\;crit}=\frac{1}{2}MV_{CIV}^{2}$$

The analysis of the experiments described in this paper starts from the energy balance. Equation (4) expresses the energy balance formulated by Grady et al. [14], which described the HVI of cylindrical brittle fragments on steel targets.
(4)$$\frac{1}{2}MV_{0}^{2}=\frac{1}{2}\left(M_{res}+M_{plug}\right)v_{res}^{2}+E_{x}$$
where M is the initial mass of the fragment, $M_{res}$ and $M_{plug}$ are, respectively, the residual mass of the fragment and the mass of the plug ejected from the plate; $V_{0}$ indicates the initial velocity and $E_{x}$ was defined by Grady et al. [14] as “excess energy”, which was expressed as the following sum
(5)$$E_{x}=E_{res\;frg}^{kin}+W_{p}+E_{f}$$
where the value $E_{res\;frg}^{kin}$ is the residual kinetic energy associated with the expansion of the shattered fragment; *W_p_* is the energy dissipated by the plate in the perforation process, and *E_f_* is the energy absorbed by the fragment for shattering. The balance can be simplified by imposing the *CIV* as the initial velocity in Equation (4): as result, the residual kinetic energy becomes null. Therefore, by combining Equations (3)–(5), the balance can be rewritten as follows
(6)$$E_{Kin\;crit}=\frac{1}{2}MV_{CIV}^{2}=W_{p}+E_{f}$$

Further simplification of the balance can be made following the observations of Grady et al. [14], which remarked that the term *E_f_* is negligible for brittle materials, as they experience fracture without any significant plastic deformation and, therefore, without or with minor energy dissipation. Consequently, *E_Kin Crit_* can be approximated to *W_p_*, as reported in Equation (7).
(7)$$E_{Kin\;crit}=\frac{1}{2}MV_{CIV}^{2}=W_{p}+E_{f}≅W_{p}$$

The quantitative evaluation of the term *W_p_* has been the subject of intensive study. Empirical equations for the evaluation of the term *W_p_* were defined in the forties by Bethe [15] and later by Taylor [16] and have been continiously improved over the years as summarised by Rosenberg et al. [10]. In this work an empirical formula, shown in Equation (8), is used for the evaluation of the term *W_p_*
(8)$$W_{p}=\pi k\sigma_{u}dh^{2}$$
where *d* is the diameter of the fragment, *h* is the thickness of the plate, and $\sigma_{u}$ is the ultimate stress of the plate, multiplied by the constant *k*, dependent on the strain rate. Historically, the term k represented a multiplier of the strength of the plate obtained experimentally, dependent on the mechanical and geometrical features of fragments and plates involved. Recent studies such as the investigations discussed by Meyer et al. [17] and Stepanov [18] provide a physical explanation to the need of a multiplier for the ultimate strength $\sigma_{u}$ during a dynamic impact. In particular, the study performed by Meyer et al. [17] discusses the increase of yield strength observed in metals during very high strain rate loadings ($\dot{\varepsilon }\geq 10^{5}-10^{6}\;s^{-1}$). Stepanov [18] also investigated the influence of strain rate of impact and explosive loading conditions on the mechanical properties of high strain steels, observing that for $\dot{\varepsilon }\geq 10^{5}s^{-1}$ the strength values exceed the static value several times. The strain rates characterising the HVI discussed in this work are estimated to be in the ranges of 4 ÷ 5 * 10^5^ s^−1^, and, therefore, the term *k* in Dquation (8) is associated with the significant increase of yield strength experienced by the armour steel plates during the impacts. The estimated strain rate values were obtained by dividing the impact velocities by the thickness of the plate, as discussed by Cagle et al. [8]. The values obtained align with indications by Zukas [9].

By combining Equation (7) with the formula in Equation (8), the balance could be expressed as follows
(9)$$E_{Kin\;crit}=\frac{1}{2}MV_{CIV}^{2}=W_{p}+E_{f}≅W_{p}=\pi k\sigma_{u}dh^{2}$$

This last form of the energy balance could be rearranged through algebraic manipulation to make it adimensional, similarly to what was done by Aly et al. [19]. In the paper, seven different empirical equations valid for predicting $E_{Kin\;crit}$ for cylindrical and hemispherical fragments impacting metallic plates were compared. The different equations were expressed in adimensional form to identify the non-dimensional parameters affecting the normalised $E_{Kin\;crit}$. It was observed that the parameter (*h/d*), where *h* is the plate thickness and *d* is the diameter of the fragment, plays a critical role in evaluating the energy necessary for perforation.

Similarly to what was described by Aly et al. [19], the energy balance in Equation (9) was rewritten in adimensional form as a function of the parameter (h/d)
(10)$${\tilde{E}}_{kin}^{crit}\left(\frac{h}{d}\right)=\frac{E_{Kin\;crit}}{\sigma_{u}d^{3}}=\frac{W_{p}}{\sigma_{u}d^{3}}+\frac{E_{f}}{\sigma_{u}d^{3}}=\varepsilon *{\left(\frac{h}{d}\right)}^{2}+\varphi *\left(\frac{h}{d}\right)≅\varepsilon *{\left(\frac{h}{d}\right)}^{2}$$
where ${\tilde{E}}_{kin}^{crit}$ represents the normalised $E_{Kin\;crit}$. It can be observed that the parameter ε represents the following expression
(11)ε=π∗k
which allows estimating the value of the multiplier *k*. 

The analytical model in Equation (10) was used to fit the ${\tilde{E}}_{kin}^{crit}$ calculated from the experimental data. The expression shows that ${\tilde{E}}_{kin}^{crit}$ is equal to a two-term quadratic equation. However, according to Grady et al. [14], the linear term is negligible in first approximation. An assessment of the validity of the approximation, the estimation of the parameters ε and φ from Equation (10) and *k* from Equation (11) are described in Section 4.

## 3. Results

Table 2 lists the experimental outcomes for the MRE and steel fragments impacting armour steel plates with different thicknesses. The results are categorised in function of the characteristic *h/d* parameter, where *h* represents the plate thickness and *d* the diameter of the fragment impacting the target. The outcomes of each shot in termos of perforation are indicated with X and V for non perforation and perforation of the targets, respectively.

### 3.1. Critical Impact Velocity (CIV), Critical Impact Energy ($E_{Kin\;crit}$
) and Normalised $E_{Kin\;crit}$

Table 3 lists the *CIV*, the $E_{Kin\;crit}$, $E_{Kin\;crit}^{Specific}$ and the ${\tilde{E}}_{kin}^{crit}$ defined using the shots listed in Table 2.

The data listed in Table 3 show that the *CIV* increases with increasing plate thickness for all the tested configuration. The *CIV* values for 4.6 mm steel fragments (yellow markers) are in the range of 20 ÷ 28% lower than the values obtained for 5 mm MRE fragments impacting targets with the same thickness, and in the range of 39 ÷ 45% lower than the values obtained for 3.5 mm MRE fragments impacting the same targets.

The same trend is reflected on the $E_{Kin\;crit}$, $E_{Kin\;crit}^{Specific}$ and the ${\tilde{E}}_{kin}^{crit}$ defined. The $E_{Kin\;crit}$ outcomes are plotted in Figure 6. Results for 5 mm MRE fragments and 4.6 mm steel fragments lie on the same kinetic energy curve as they possess the same mass. The $E_{Kin\;crit}$ values for 4.6 mm steel fragments (yellow markers) are lower (35 ÷ 49%) than the values obtained for 5 mm MRE fragments impacting targets with the same thickness.

The $E_{Kin\;crit}$ values defined for the 3.5 mm MRE fragments are lower than the values obtained for 5 mm MRE fragments impacting the same target (52% for 2 mm plates and 30% for the 3 mm plates).

On the othe hand, when compared with the steel fragments, the values of $E_{Kin\;crit}$ estimated for the 3.5 mm MRE fragmnets are similar, (circa 8% difference). Obviously, the fact that the 3.5 mm MRE fragments weigh 1/3 than the steel fragments is reflected on the *CIV*, resulting in the translation of the points along the x axis.

Figure 7 shows the $E_{Kin\;crit}$ per unit mass for all the configurations tested. In this case, the $E_{Kin\;crit}^{Specific}$ values for the MRE fragments of 3.5 and 5 mm follow the same trend, while the values for the steel fragments are between 40 and 50% lower.

Furthermore, the graph indicates that the $E_{Kin\;crit}^{Specific}$ values observed for 3.5 mm MRE fragments are higher than the respective values obtained for 5 mm fragments. The difference magnifies increasing *h/d*, going from 16% for the 2 mm plates to 25% for 2.5 mm plates and 30% for the 3 mm plate.

The graph in Figure 8 depicts the ${\tilde{E}}_{kin}^{crit}$ over *h/d*. The lower energy values obtained are linked with the different failure modes observed: MRE are brittle and undergo fragmentation upon impact, while the steel fragments are ductile and deform plastically, as visible in Figure 9. The failure mode influences significantly the fragment/plate interaction, resulting in lower impact velocities required to perforate the plates using steel fragments. This is reflected in values of ${\tilde{E}}_{kin}^{crit}$ between 28 and 48% lower for steel fragments are than for the MRE fragments.

The frames in Figure 9 compare the time evolution of two impacts involving, on the left, a 4.6 mm steel fragment impacting a 2 mm plate at 693 m/s, and, on the right, a 5 mm MRE fragment impacting a 2 mm plate at 877 m/s.

Figure 9c displays the deformed steel fragment, having the characteristic “mushroom” shape. Figure 9f, on the other hand, shows the shattered MRE fragment and IIER. This brittle nature of MRE leads to fragmentation even at low impact velocities. The impact loadings cause intense stresses and internal friction, resulting in fragmentation and energy release. The shattered MRE fragment continues its trajectory behind the perforated plate while expanding as a debris cloud, essentially following the launch trajectory.

The high-speed recordings showed that the shattered parts of the fragment travel with comparable residual velocities of the plugs, provided the absence of rotations observed for impact velocities close to CIV. It was not possible to recover the shattered fragment as, continuing its trajectory, it further reacted, impacting walls of the chamber. 

IIER was observed in the form of light emission for all experiments involving MRE. Figure 9f shows the energy release spreading. It can be observed that the energy release front travels with the same velocities of the shattered fragment and the plug.

### 3.2. Plate Perforation

The primary failure mode observed for the targets impacted at velocities above the *CIV* was shear plugging for both the 3.5 mm and the 5 mm MRE fragments, as can be onserved in Figure 10. Shear plugging is the typical failure mode for materials with high strength and high failure strains, such as armour steel, as discussed by Hazell [16] for instance, who concluded that plugging is more favourable than plate deformation.

The cross-sections of the plates show that the perforation channels have diameters equal to the diameter of the impacting fragments. In the case of thicker plates, shown in Figure 10a,b, minor hole enlargement on the impact surface was observed.

In Figure 10c,d, some minor dishing was observed in 2 mm plates impacted by both the 3.5 mm and 5 mm MRE fragments at velocities close to the *CIV*.

The failure mode of the plate influences directly the perforation process and, consequently, the post-perforation IIER spread behind the plate. Details on the IIER velocity will be discussed in detail in Section 3.3 and Section 3.4.

### 3.3. Residual Velocity Analysis

The RI curves for the experimental results were determined. Quantitative considerations are discussed for the firings involving the 5 mm MRE fragemnts, as the experimental setup does not allow to reach velocities greater than 1600 m/s. 

In the case of 3.5 mm MRE fragments, it was observed that, due to the bluff shape and the mass of only 0.2 g, the flight stability of was severely affected for velocities higher than 1200 m/s, and, therefore, the data collected are not sufficient for a reliable curve fit, as observable in Figure 11.

The results from Table 4 are plotted in Figure 12. The RI fitting coefficients *a* and *b* are defined for each case through a least square fit of the experimental data.

The RI curves indicate that the increase of thickness of the plates from 2 mm to 2.5 mm results in residual velocities circa 20% lower, while the increase from 2.5 mm to 3 mm plates results in a decrease of circa 30%, indicating an average increase of 5% in velocity loss per additional millimetre.

Figure 13 shows the kinetic energy loss relative to the shots depicted in Figure 12. The experimental distributions depicted in Figure 13 follow a linear trend for each fragment/plate combination, as highlighted by the curve fit indicating an increase of circa 10% in terms of kinetic energy loss with increasing plate thickness by 5 mm, i.e., 2% circa per additional millimetre.

### 3.4. Impact Induced Energy Release

The experimental set-up used to record the IIER did not allow to measure the impact angles for shots reported in Table 5. However, the residual velocities recorded are coherent with the analyses discussed in this paper. 

The evaluation of the energy release front velocities, summarised in Table 5 shows that the energy release travels with velocities comparable to the residual velocities of shattered fragments and plugs flying without tumbling, as shown in Figure 3.

A visual comparison of firings involving 5 mm MRE fragments impacting 2.5 mm target at velocities of 1252 m/s and 1220 m/s are depicted in Figure 14, respectively, on the left and the right. The frames are taken with the same time intervals.

The velocity difference between the two shots is 2.5%, which is sufficiently close for a qualitative discussion.

The frames on the left, recorded in a dark room, highlight the IIER time evolution, while the frames on the right show the mechanical aspects of plate perforation and MRE fragment shattering. 

In Figure 14a, the first reaction is triggered upon impact. The process continues and lasts approximately 30 µs until the plug is ejected from the target and the energy release is visible behind the target, as indicated in Figure 14c. In Figure 14d–f he energy release continues spreading behind the plate, mainly following a normal direction. The post-perforation energy release lasts approximatively 600 µs, as indicated by Figure 14g.

A significant amount of energy release is observed prior to perforation. During the perforation, the MRE samples are shattered into numerous smaller parts, as visible in Figure 14e,f. The reaction continues spreading and expands, following a conical shape.

## 4. Discussion

Figure 15 displays the values of ${\tilde{E}}_{kin}^{crit}$ derived from the experimental results for each fragment-plate combination over the *h/d* parameter. The trend shown by the experimental ${\tilde{E}}_{kin}^{crit}$ values in follow the same tendency observed by Børvik [20] and Wen [21] in their papers describing the transition zone between shear plugging and adiabatic shear plugging. In both publications, the transition zone between plugging failure modes was identified for values of *h/d* ranging between 0.5 and 0.6. 

As reported by Equation (10), the value of ${\tilde{E}}_{kin}^{crit}$ is expressed by a two-term quadratic equation but, following the indications of Grady et al. [14], the linear term of the equation can be neglected and ${\tilde{E}}_{kin}^{crit}$ is then approximated by a one-term equation. The fitting parameters determined through a least square fit from the experimental data are listed in Table 6. It can be observed that, in the two-terms equation, the linear parameter φ is one order of magnitude smaller than ε. 

The estimated values of the multiplier *k* are sensible. Preliminary Taylor impact tests have been performed and the results indicate the validity of the estimation. Furthermore, the estimations align significantly with the observations of Meyer et al. [17] and Stepanov [18] discussed in Section 2.3. Further details on the dynamic experimental and numerical characterization of MRE are described in a dedicated publication by the authors of this paper [22].

The graph in Figure 15 shows the two curves, respectively, in red and black. Both curves are accurate for values of (h/d)≥0.57, while a certain degree of discrepancy can be observed for lower ranges of (h/d), as indicated by the graph. The red curve is still sufficiently accurate at the *h/d* value of 0.5, with an error of only 2%, while the black curve shows an error in the order of 10% at the same abscissa. The inaccuracy becomes more significant at *h/d* of 0.4, reaching a value of 30% for the one-term form of the model.

Overall, the energy-based model is significantly reliable, giving accurate indications on the ${\tilde{E}}_{kin}^{crit}$. By neglecting the linear term of the model, the evaluations are still significantly accurate, indicating that the observations formulated by Grady et al. [14] can be applied to this case study.

It is worth noting that the energy-based model does not consider the physical aspects involved in the perforation process at the base of observations, as Børvik et al. [23] pointed out. In fact, Wen et al. [21] observed that the failure modes of plates impacted by blunt projectiles are dependent on the parameter *h/d*. In particular, in the area ranging from values of *h/d* of 0.3 to 0.5, the plates fail by simple shear plugging. As discussed in Section 3.2, minor dishing was observed in the area surrounding the craters in this zone. The deformation of the plate increases the energy dissipation and is reflected in the ${\tilde{E}}_{kin}^{crit}$ trend observed. The area going from values of h/d≥0.6, indicates the adiabatic shear plugging zone. Adiabatic Shear Bands (ASB) formation plays a crucial role in forming the plug, resulting in lower energies necessary for the perforation. The higher impact velocities necessary for perforation in this zone are responsible for the adiabatic shear plugging initiation [21]. The area ranging from *h/d* of 0.5 to 0.6, represents the transition zone. The subdivision just described represent a reference accepted by the scierntific community and perfectly alignes with the observations and conclusions of Børvik et al. [20,23] and Wen et al. [21]. However, a clear boundary between shear plugging, transition and adiabatic shearing cannot be defined, as the formation of ASB is an evolutionary process that starts from, approximately, 0.5 *h/d* and is strongly affected by the variability of microstructural morphology that is intrinsic in every commercial product, as observed by Xu et al. [24], by Couque [25], Yiadom et al. [26] or Jo et al. [27].

Metallographic analyses were performed on sections of the perforated plates to validate the observations discussed in this chapter. Figure 16 shows the presence of ABS in a 3 mm plate impacted by a 5 mm MRE fragment at 1141 m/s.

## 5. Conclusions

94 ballistic impact experiments were performed to collect relevant information on the behaviour of MRE fragments impacting armour steel plates with thicknesses ranging from 2 mm to 3 mm. MRE samples are used as surrogate material of IRM, considering the affinity in terms of mechanical properties and macroscopic IIER effects. Quantitative parameters as *CIV*, $E_{Kin\;crit}$, ${\tilde{E}}_{kin}^{crit}$ and residual velocities were evaluated.

Prior to the set-up of the experiments, the influence of the pitch angle (α) on the impact outcomes was assessed. In particular, it was observed that for values of α up to 25°, the residual velocities measured were not affected significantly, as shown by the Recht-Ipson curves in Section 2.3.

The MRE fragments shattered upon impact, continuing the trajectory behind the perforated target plates. The comparison with inert steel fragments led to the conclusion that the failure mode influences the *CIV* and $E_{Kin\;crit}$ quantification significantly, resulting in lower values of both CIV and $E_{Kin\;crit}$ for steel fragments. It was concluded that by increasing the thickness of the plates by 0.5 mm, the kinetic energy loss increases by 10%.

The failure mode observed in the plates is an indication of the energy necessary to perforate the plate. The interaction between the target plates and the MRE fragments at impact velocities greater than the *CIV* resulted in shear plugging failure for all fragment/plate combinations. However, for lower values of *h/d*, simple shear plugging was preceded by minor dishing, while for the higher values of *h/d* adiabatic shear plugging was observed, affecting the *CIV* and $E_{Kin\;crit}$.

Impact induced energy release was observed for all shots performed using MRE fragments. The IIER front spreads with velocities comparable with the residual velocities of the shattered fragment and, provided a stable flight, the ejected plugs.

The energy-based model developed is sufficiently accurate for the ${\tilde{E}}_{kin}^{crit}$ evaluation for observed. The comparison between the two-terms form and the one-term form of the model indicates that the energy absorbed by the fragment for shattering is negligible, at least in first approximation, due to the brittle nature of MRE, confirming the assumptions formulated by Grady et al. [14].

## Figures and Tables

**Figure 1 materials-14-04649-f001:**
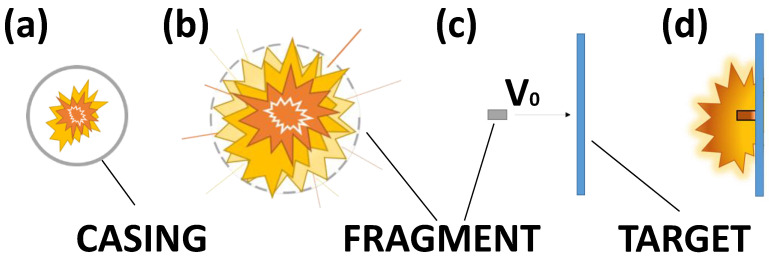
Schematic representation of bomb detonation (**a**); case fragmentation (**b**); fragment flight accelerated by the blast (**d**); impact with target and impact-induced energy release (**c**).

**Figure 2 materials-14-04649-f002:**
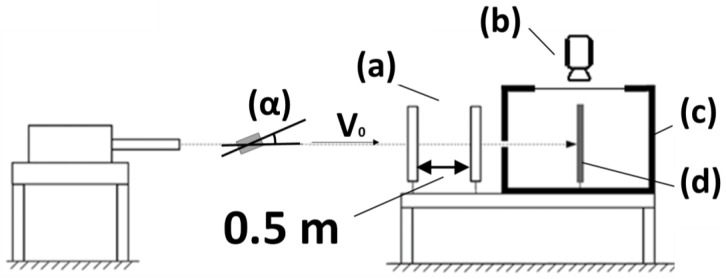
Ballistic experiment schematic set-up: (**a**) light barrier; (**b**) high-speed camera; (**c**) ballistic chamber; (**d**) target; (**α**) impact-pitch angle.

**Figure 3 materials-14-04649-f003:**
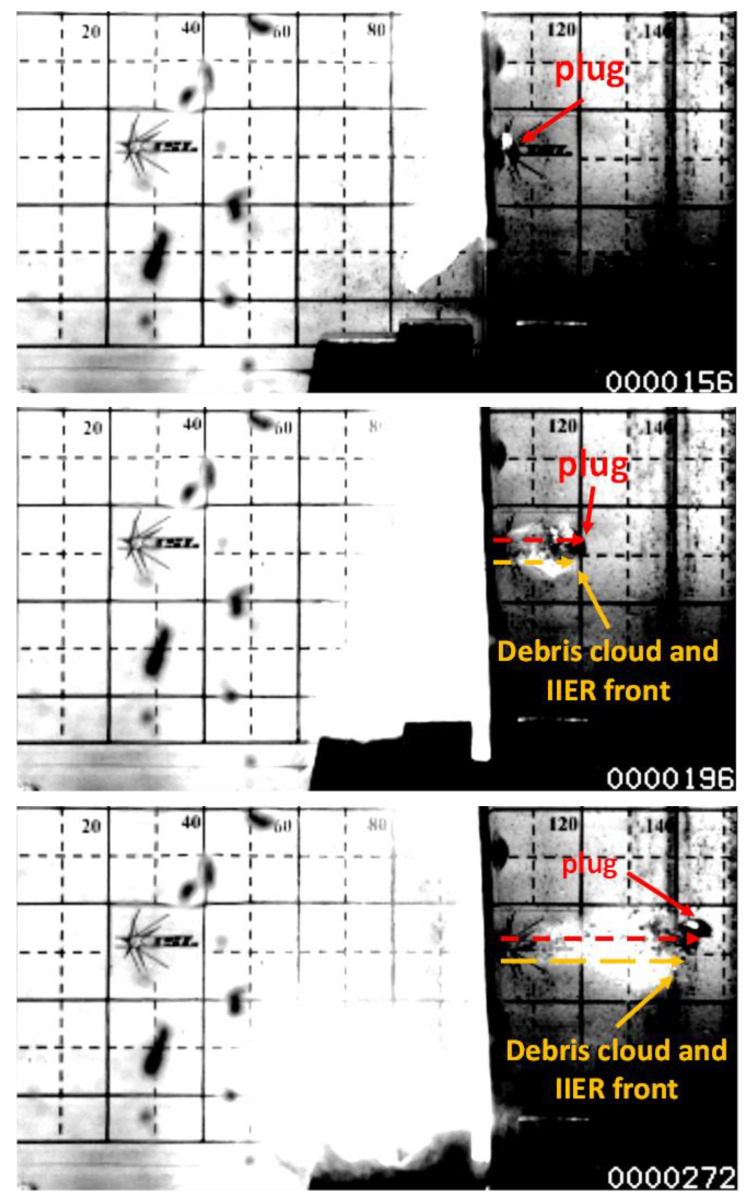
5 mm MRE fragment impacting 2 mm plate at 889 m/s. Estimated residual velocity of 325 m/s.

**Figure 4 materials-14-04649-f004:**
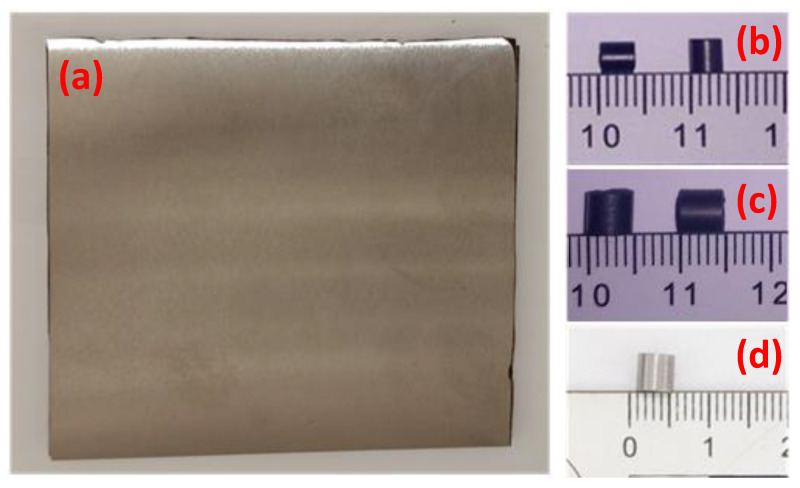
(**a**) 10 mm × 10 mm × 2 mm armour steel plate; (**b**) 3.5 mm MRE fragments; (**c**) 5 mm MRE fragments; (**d**) 4.6 mm steel fragment.

**Figure 5 materials-14-04649-f005:**
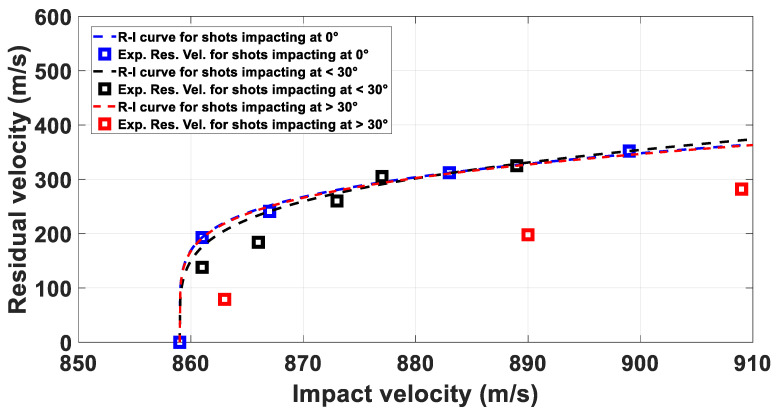
Recht Ipson curves for 5 mm fragments impacting 2 mm plates.

**Figure 6 materials-14-04649-f006:**
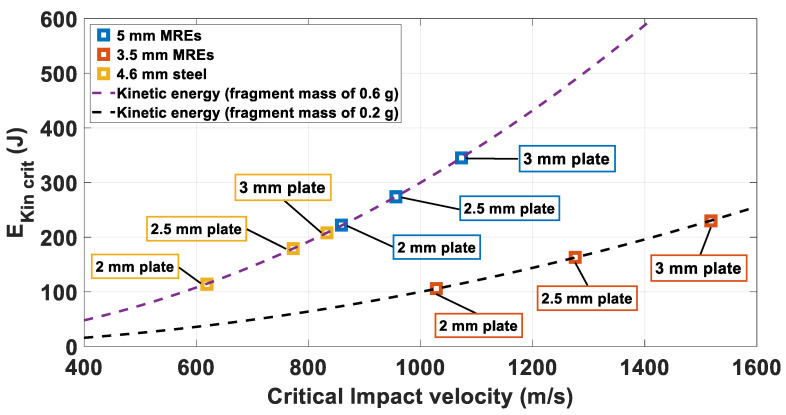
Critical kineti energy over critical impact velocities. The dashed lines indicate the kinetic energies for fragments having mass of 0.6 and 0.2 g, respectively in purple and black, as function of impact velocity.

**Figure 7 materials-14-04649-f007:**
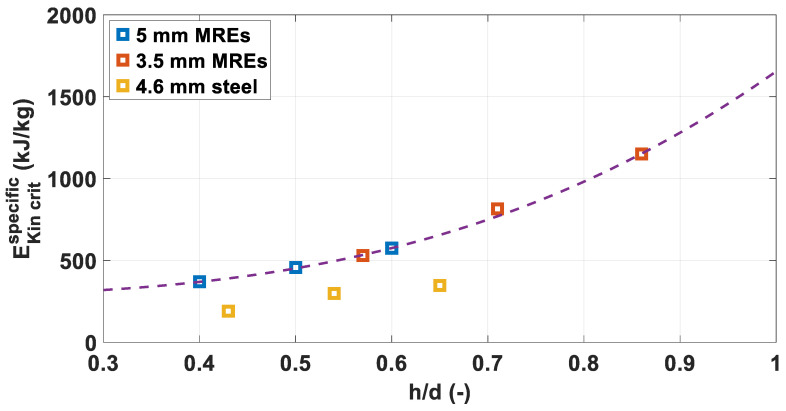
Specific Critical Impact Energies over normalised plate thickness. The dashed line indicates the kinetic energy per unit mass.

**Figure 8 materials-14-04649-f008:**
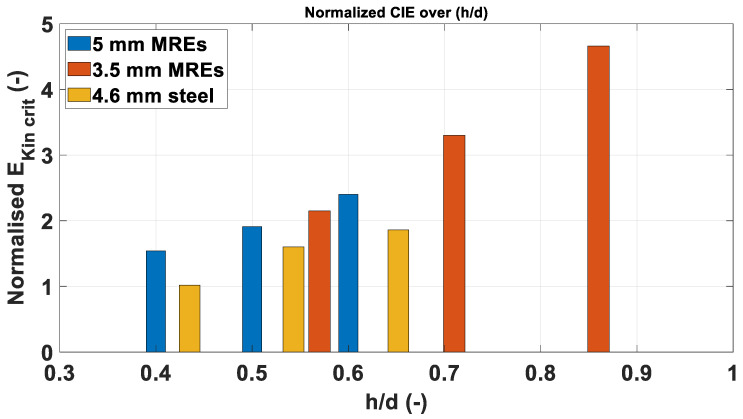
Plot of estimated values of ${\tilde{E}}_{kin}^{crit}$ over normalised plate thickness.

**Figure 9 materials-14-04649-f009:**
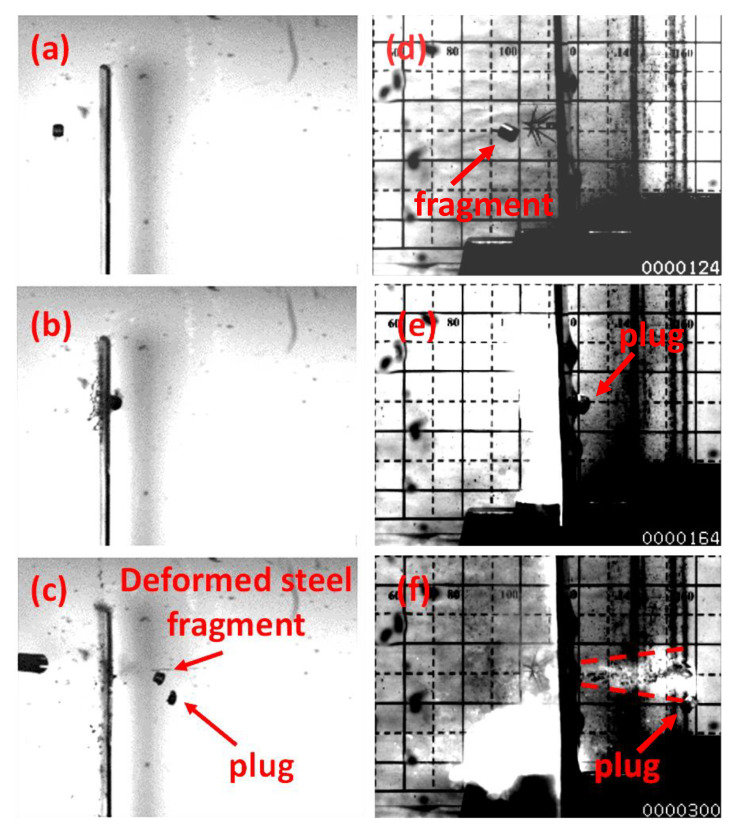
Failure mode comparison. On the left, 4.6 mm steel fragment impacting 2 mm plate at 693 m/s; on the right, 5 mm MREs fragment impacting 2 mm plate at 877 m/s. (**a**) steel fragment flying towards the target, (**b**) impact between steel fragment and the plate, (**c**) the deformed fragment perforates the target and the plug is ejected; (**d**) MRE fragment flying towards the target with 24 degrees of yaw, (**e**) impact, pre-perforation energy release and plug formation, (**f**) the fragment shatters during the interaction with the target and spreads behind the plate.

**Figure 10 materials-14-04649-f010:**
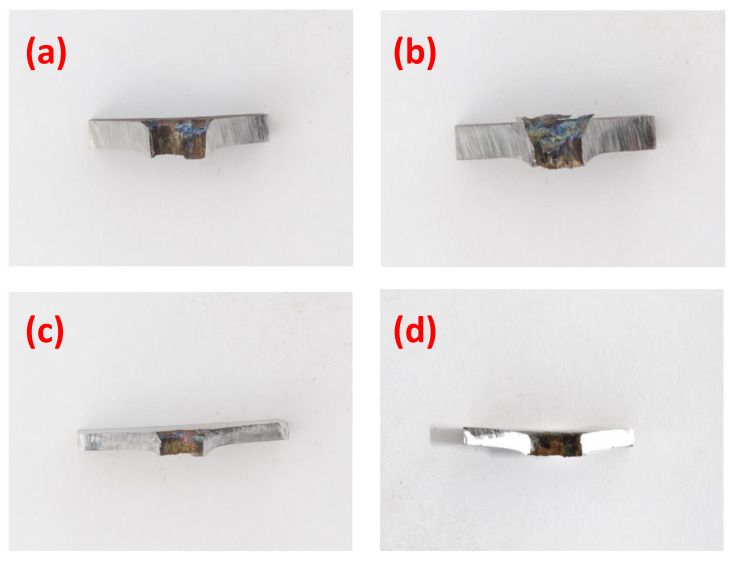
Cross sections of perforated plates. In detail: (**a**) 3 mm plate impacted by 5 mm fragment at 1075 m/s; (**b**) 3 mm plate impacted by 3.5 mm fragment at 1549 m/s; (**c**) 2 mm plate impacted by 5 mm fragment at 861 m/s; (**d**) 2 mm plate impacted by 3.5 mm fragment at 1023 m/s.

**Figure 11 materials-14-04649-f011:**
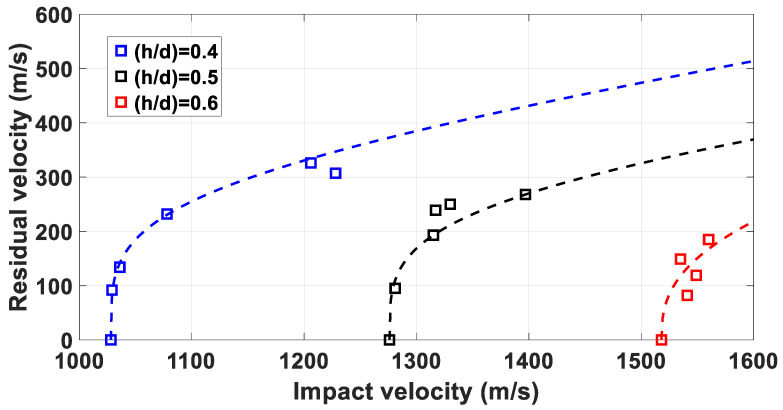
Recht Ipson curve fittings for 3.5 mm fragments: the blue markers indicate residual velocities observed impacting 2 mm plates; the black markers indicate residual velocities measured impacting 2.5 mm plates and the red markers indicate residual velocities recorded impacting 3 mm plates.

**Figure 12 materials-14-04649-f012:**
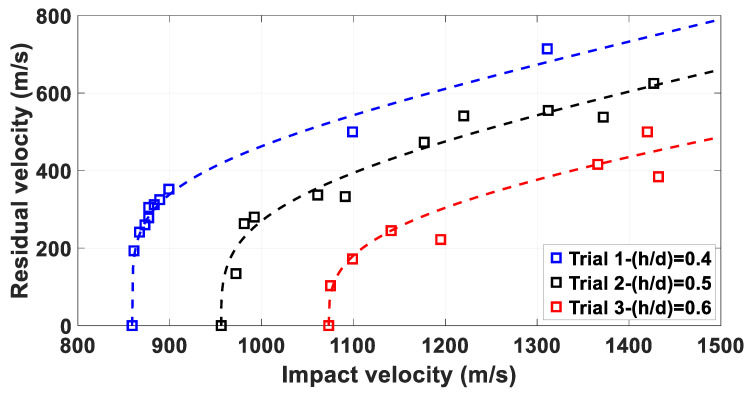
Recht Ipson curve fittings for 5 mm fragments impacting targets with values of α lower than 30°.

**Figure 13 materials-14-04649-f013:**
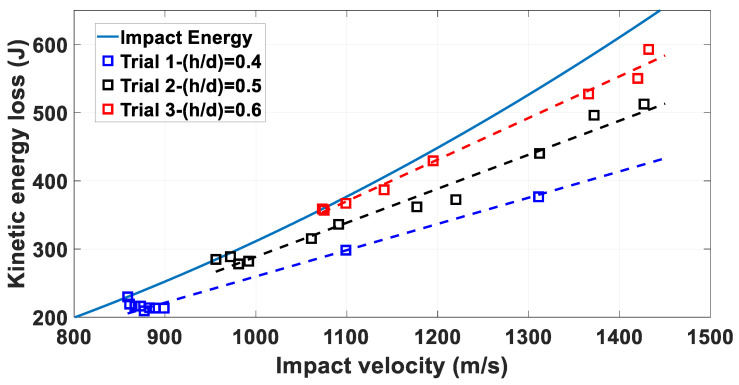
Kinetic energy loss for 5 mm fragments over impact velocity. The blue markers indicate the amout of kinetic energy lost impacting 2 mm plates; the black markers indicate the kinetic energy loss observed when impacting 2.5 mm plates and the red markers indicate the kinetic energy loss measured impacting 3 mm plates. The dashed lines are linear curve fit obtained from the experimental data.

**Figure 14 materials-14-04649-f014:**
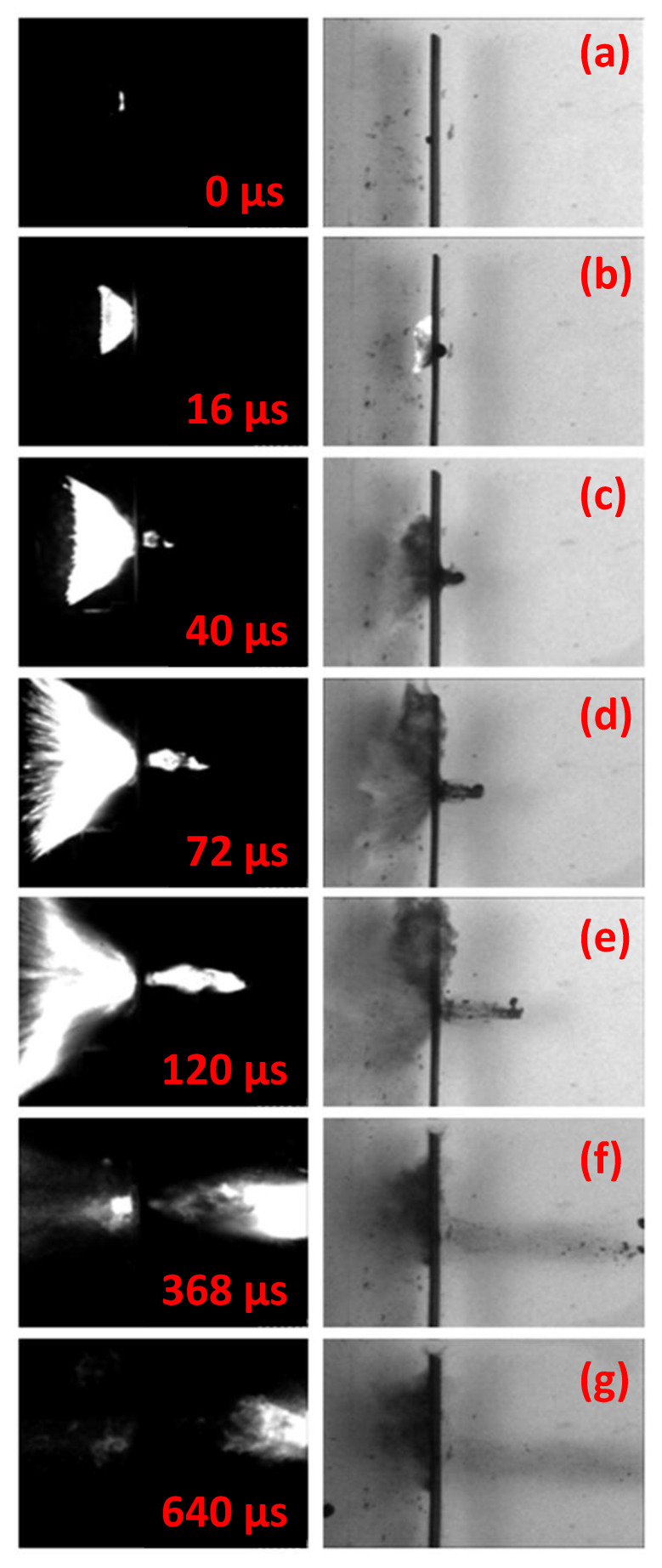
Time evolution of high-velocity impact for 5 mm MRE fragments impacting 2.5 mm plates. Left: energy release at impact velocity 1252 m/s. Right: impact velocity 1220 m/s. (**a**) impact between fragment and target, (**b**) interaction between the fragments and the targets and plug formation; (**c**) plug ejection; (**d**–**g**) plug flight behind the plate, debris cloud spreading and post-perforation energy release evolution.

**Figure 15 materials-14-04649-f015:**
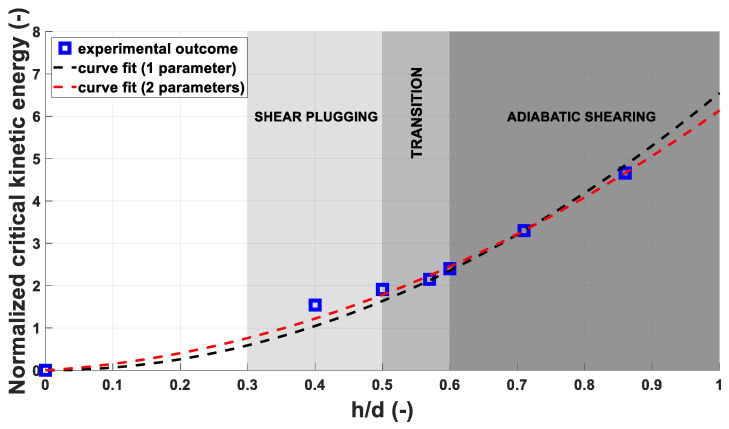
${\tilde{E}}_{kin}^{crit}$ over *h/d* for every fragment-plate combination.

**Figure 16 materials-14-04649-f016:**
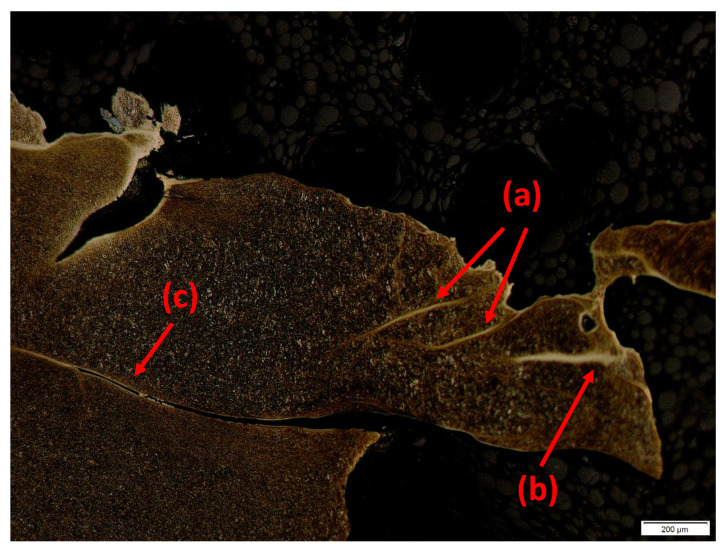
Metallographic analysis of plate impacted at 1141 m/s with an h/d = 0.6; (**a**) details of wide dASBs and gradual formation of tASB; (**b**) deformed shear bands (dASB), which are characteristic for early-stage deformations and shear; (**c**) narrow transformed shear band (tASB) created by further development of the dASB.

**Table 1 materials-14-04649-t001:** Mechanical properties of MRE [6] and armour steel.

Material	E (GPa)	Yield Str. (MPa)	Ult. Str. (MPa)	ρ (g/cm^3^)
**MRE**	15.9 ÷ 31.7	96.5 ÷ 126.3	128.9 ÷ 159.3	6.35
**Armour steel**	210	1050	1150	7.50

**Table 2 materials-14-04649-t002:** Summary of experimental results.

Material	*h* (mm)	*d* (mm)	*h/d*	Mass (g)	α° (deg)	V_0_ (m/s)	$v_{res}$ (m/s)	$\Delta E_{kin}$ (J)	Perforation
**MRE**	2	5	0.4	0.6	16	843	0	221	X
					2	852	0	226	X
					22	857	184	141	V
					19	861	0	231	X
					0	861	193	139	V
					0	867	241	122	V
					0	870	0	236	X
					12	873	260	117	V
					24	877	278	112	V
					18	877	305	102	V
					0	883	312	102	V
					28	889	325	99	V
					0	899	352	93	V
					2	1099	500	112	V
					18	1311	714	111	V
**MRE**	2.5	5	0.5	0.6	0	883	0	243	X
					8	909	0	257	X
					4	928	0	268	X
					10	938	153	192	V
					7	960	0	287	X
					10	972	134	219	V
					22	981	263	161	V
					20	992	280	160	V
					20	1061	337	163	V
					2	1091	333	179	V
					20	1177	473	154	V
					28	1220	541	144	V
					ND ^1^	1252	-	-	V
					ND ^2^	1312	555	178	V
					ND ^2^	1372	538	217	V
					ND ^2^	1427	625	200	V
MRE	3	5	0.6	0.6	18	1035	0	334	X
					16	1043	0	339	X
					5	1051	0	344	X
					23	1060	0	350	X
					0	1075	103	294	V
					10	1085	214	236	V
					5	1099	172	268	V
					12	1141	245	250	V
					22	1366	416	281	V
					ND ^1^	1420	500	264	V
					ND ^1^	1432	384	342	V
MRE	2	3.5	0.57	0.2	0	985	0	103	X
					23	997	0	106	X
					0	1023	87	93	V
					0	1029	98	92	V
					0	1034	0	114	X
					12	1036	134	87	V
					28	1045	0	117	X
					26	1078	232	77	V
					0	1206	326	77	V
					27	1228	307	85	V
MRE	2.5	3.5	0.71	0.2	26	1240	0	164	X
					0	1245	0	166	X
					17	1248	31	158	V
					20	1270	0	173	X
					22	1281	95	150	V
					28	1290	0	178	X
					28	1315	193	135	V
					25	1317	239	124	V
					30	1330	250	117	V
					28	1397	268	127	V
MRE	3	3.5	0.86	0.2	3	1491	0	238	X
					19	1495	0	239	X
					8	1502	0	241	X
					6	1508	149	198	V
					15	1532	0	251	X
					29	1541	82	228	V
					28	1549	119	219	V
					7	1560	185	202	V
Steel	2	4.6	0.43	0.6	0	550	0	91	X
					2	583	0	102	X
					0	597	75	82	V
					3	600	0	108	X
					10	630	142	71	V
					0	643	134	78	V
					5	656	0	129	X
					0	693	154	87	V
Steel	2.5	4.6	0.54	0.6	5	733	0	161	X
					12	735	0	162	X
					5	771	91	139	V
					0	773	0	179	X
					4	777	135	124	V
					0	785	174	112	V
					11	806	189	114	V
					2	808	0	196	X
Steel	3	4.6	0.65	0.6	10	786	0	185	X
					3	806	104	148	V
					0	824	0	204	X
					0	831	167	132	V
					18	833	0	208	X
					0	840	0	212	X
					8	852	176	137	V
					2	866	182	140	V

^1^ Shot performed in the darkroom. The angle of impact and residual velocities were not quantifiable. ^2^ Shot performed in the darkroom. The angle of impact is not quantifiable.

**Table 3 materials-14-04649-t003:** Summary of results for *CIV*, $E_{Kin\;crit}$ and ${\tilde{E}}_{kin}^{crit}$ evaluated from shots listed in Table 3.

Material	*h/d*	*CIV* (m/s)	$E_{Kin\;crit}$ (J)	$E_{Kin\;crit}^{Specific}$ (kJ/kg)	${\tilde{E}}_{kin}^{crit}\left(\frac{h}{d}\right)$
MRE	0.4	859	222	370	1.54
	0.5	956	274	457	1.91
	0.6	1073	345	575	2.40
MRE	0.57	1028	106	530	2.15
	0.71	1276	163	815	3.30
	0.86	1518	230	1150	4.66
Steel	0.43	619	114	190	1.02
	0.54	773	179	298	1.60
	0.65	833	208	347	1.86

**Table 4 materials-14-04649-t004:** Summary of parameters for the RI equation.

(*h/d*)	CIV (m/s)	*a*	*b*
**0.40**	859	0.5358	4.642
**0.50**	956	0.4713	3.473
**0.60**	1073	0.3712	3.166

**Table 5 materials-14-04649-t005:** Summary of parameters for the RI equation.

*h/d*	Shot	CIV (m/s)	*v_res plug_* (m/s)	*v_energy release front_* (m/s)
0.5	14	1312	555	555
	15	1372	538	538
	16	1427	625	625
0.6	10	1420	500	500
	11	1432	384	416

**Table 6 materials-14-04649-t006:** Summary of parameters defined for the ${\tilde{E}}_{kin}^{crit}$ equation.

	ε	φ	*k*
*Two-term eq.*	5.14	0.99	1.6
*One-term eq.*	6.55	-	2.1

## Data Availability

Data is contained within the article.

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
