# Peer review of "High-Velocity Impacts of Pyrophoric Alloy Fragments on Thin Armour Steel Plates"

_materials, 2021, doi:10.3390/ma14164649_

Round 1

Reviewer 1 Report

This paper address the RM impact condition with in-depth analytical calculation. 

This reviewer sees enough scientific depth on this study and approach, however, due to the lack of literature review in the early part of the paper, I was having hard time to see the big picture.  In fact, the necessary literature reviews were included in the analysis section just before the introduction of equations without much in-depth description, and I ended up with questions. 

For example, many of analytical equations are from other publications, and I saw several equations based on static values (i.e. young's module, yield strength, ultimate strength, etc.), then, how these values will affect the high velocity dynamic event that is the main point of this paper?

Have you done any kind of impact driven shock analysis?

In general, I felt that this paper has some scientific values and it would be more readers friendly if more literature reviews are added. 

Author Response

Dear reviewer,

I would like to express my gratitude to you for taking the time to read and assess my work, providing relevant questions. In the following section I address the concerns raised point by point. I am uploading a new version of the manuscript, updated according to the suggestions of the reviewers. The updated parts are written in green.

Thank you again for taking the time to review the work.

Kind regards, 

Evaristo Santamaria Ferraro.

Response to Reviewer 1 Comments

Point 1: many of analytical equations are from other publications, and I saw several equations based on static values (i.e. young's module, yield strength, ultimate strength, etc.), then, how these values will affect the high velocity dynamic event that is the main point of this paper?

Response 1: 
Page 8 lines 244 onwards.
The equations listed in the paper represent the starting point of the analytical discussion. Each formula represents a relation derived from extensive experimental analyses by the authors and represents a well-established analytical tool in the ballistic community, as shown by Zukas [1] and Rosenberg et al. [2], just to name a few.
For instance, equation 8 in page 7
W_p=πkσ_u dh^2,    (8)
Represents the evaluation of the energy dissipated by the plate in the perforation process. As stated, “The quantitative evaluation of the term Wp has been the subject of intensive study. Empirical equations for the evaluation of the term Wp were defined in the forties by Bethe [3] and later by Taylor [4] and have been continuously improved over the years as summarised by Rosenberg et al. [2]”.
In the equation the term k represents a multiplier of the strength of the plate. Recent studies, performed by Meyer et al. [5] and Stepanov [6] provide a physical explanation to the need of a multiplier for the ultimate strength σ_u during a dynamic impact. In particular, the authors observed that “for ε ̇≥〖10〗^5 s^(-1) the strength values exceed the static value several times”. “…therefore, the term k in equation (8) is associated with the significant increase of yield strength experienced by the armour steel plates during the impacts”.
The evaluation discussed in the paper under review agrees significantly with the experimental evaluations performed by Meyer et al. [5] and Stepanov [6], providing a value of k in the same order of magnitude (in the range of 1.6÷2.1 times the value of σ_u) of the observations.
Considerations on the validity of the evaluations are discussed in Page 20 line 446-451.

Point 2: Have you done any kind of impact driven shock analysis?

Response 2: 
Page 20 lines 446-451
At the moment the material characterisation is scheduled. Taylor tests have been performed on the material. The Taylor impact test [4] was developed to evaluate the dynamic mechanical properties of materials by analysing the recovered deformed samples. However, the investigation performed by Johnson [5] highlighted that the simplifications and assumptions at the base of the Taylor approach make the methodology not adequate for a precise estimation of the dynamic properties of materials, and more accurate experimental methodologies should be preferred. Recently, Taylor tests have been used to assess deformation, fracture and damage evolution, in particular using a combination of experimental and numerical approaches. Relevant examples are the publications of Xiao et al. [6] [7], Zhang et al [8], Børvik et al. [9], or Rakvåg et al. [10] [11].
The author used the same approach due to the brittle nature of the alloy. A numerical model was developed obtaining results significantly in line with the experimental results. The work is described in the conference paper entitled ‘Investigation on Fragmentation of Pyrophoric Alloy Samples during Taylor Test Using SPH’ which will be presented at the DYMAT 13th international conference on mechanical and physical behaviour of materials under dynamic loading and published on the conference proceedings book (DYMAT format by the European Physical Journal - Web of Conference (EPJ – WoC - ISSN: 2100-014X)).
SHPB test is scheduled after the summer.
REFERENCES

1.    J. A. Zukas, High velocity impact dynamics, Wiley, 1990.
2.    Z. Rosenberg and E. Dekel, Terminal Ballistics, Springer, 2012.
3.    H. A. Bethe, “An attempt at a theory of armor penetration,” Ordonance Laboratory report R-492, 1941.
4.    G. I. Taylor, Proc. R. Soc. Lond. A, 194, (1948).
5.    G. R. Johnson, T. J. Holmquist, J. Appl. Phys., 64, (1988).
6.    X. Xiao, W. Zhang, G. Wei and Z. Mu, Mater. Des. , 31, (2010).
7.    X. Xiao, W. Zhang, G. Wei, Z. Mu and Z. Guo, Mater. Des. , 32, (2011)
8.    W. Zhang, X. Xiao, G. Wei and Z. Guo, AIP conference proceedings 1426, (2012).
9.    T. Børvik, O. S. Hopperstad, M. Langseth and K. A. Malo, Int. J. Impact Eng., 28, no. 4, (2003).
10.    K. G. Rakvåg, T. Børvik, O. S. Hopperstad and I. Westermann, EPJ Web of Conferences, vol. 26, (2012).
11.    K. G. Rakvåg, T. Børvik and O. S. Hopperstad, Int. J. Solids Struct. , 51, (2014).

Reviewer 2 Report

This paper evaluates the critical impact velocities as well as critical, specific and normalized kinetic energies. Then, the influence of impact angle on the perforation process between the fragment and an armour steel plate is discussed.

However, there are some unclear points to be reader-friendly. Some questions are listed as follows. The reviewer requires to submit the answer sheet to answer for the comments below one by one. Of course, all the comments should be reflected into the text.

  1. In Fig.2, the impact angle should be defined. Besides, the reviewer would like to know how to control the impact angle in the chamber. Perhaps, the pitch angle of target plate will be controlled by a motor.
  2. From line 129 to 133 on page #4, the authors mentioned a sabot is introduced into the apparatus. Some details about sabot including the dimensions should be shown. The complicated explanations can be found. I could not find the cartridge is used as the sabot or the sabot is used on the cartridge.
  3. On Fig. 4, the reviewer would like to know how to determine the residual velocities by the high-speed camera. Probably, it should be a magnitude. Additionally, the fragments will not perforate only one direction. The reviewer would like to know whether the camera can measure the three-dimensional components of velocity with the change in the angle after the perforation. Of course, as the authors mentioned, it is possible the fragments are shattered. Furthermore, the author mentioned “the high-speed camera recordings showed that the residual velocities of the shattered parts of the fragment are comparable with the residual velocities of the plugs and the energy release front travels with the same velocities of the shattered fragment and the plug” on Fig. 8. Detailed explanations how to measure the velocity should be given.
  4. In Fig. 4, the fitted curve cannot be drawn because CIV in Eq. (1) is unknown. The definition of CIV by Eq. (2) should be earlier.
  5. The explanations on Table 3, Fig. 7, 9, 10, 12 and 15 are insufficient. More information is required. What is the meaning on “X” and “V” in Table 3. The authors are just showing the figure and let the readers understand what is the meaning and phenomena.
  6. In Fig. 5, the result of 3.5 mm MREs should be discussed by comparing the results of 5 mm MREs as well as 4.6 mm steel. Furthermore, the details for calculation the dashed line should be clear.
  7. In Fig. 8, the figure is unclear to get sufficient information. Halation is happening in Fig. (e) and (f). Fig. (d) is so dark to identify where are the plug and fragment.
  8. On Fig. 14, the reviewer would like to know how to determine the boundary between “Shear plugging” and “Transition” as well as “Transition” and “Adiabatic shearing”.

On the minors, the following list is available.

  1. In line 134 on page #4, the reviewer cannot identify “calibre cartridge” is a proper technical word or not. The word “cartridge by caliber” can be found.
  2. In line 177 on page #5, the “red” should be replaced by “black”. In addition, 20 degree should be changed to 25 degree.

Author Response

Dear reviewer,

I would like to express my gratitude to you for taking the time to read and assess my work, providing relevant questions. In the following section I address the concerns raised point by point. I am uploading an updated version of the manuscript modified according to the suggestions of the reviewers. The modified parts are written in green.

Thank you again for taking the time to review the work.

Kind regards, 

Evaristo Santamaria Ferraro.

Response to Reviewer 2 Comments

•    Major

Point 1: In Fig.2, the impact angle should be defined. Besides, the reviewer would like to know how to control the impact angle in the chamber. Perhaps, the pitch angle of target plate will be controlled by a motor.

Response 1: Update page 3 line 107-112. Figure 2 and caption have been updated. 
The experimental observations highlighted that the fragments might fly in a certain degree of inclination due to their “bluff” shape. To the knowledge of the authors, there is no way to control the impact angle with the experimental setup used, which constitute the standard for ballistic analyses. 
This led to the necessity of investigating the influence of the angle of impact on the outcomes. The results discussed in the paper under review are significantly similar to the observations made by Bratton et al. [1], which defined a threshold impact angle of 20 degrees.

Point 2: From line 129 to 133 on page #4, the authors mentioned a sabot is introduced into the apparatus. Some details about sabot including the dimensions should be shown. The complicated explanations can be found. I could not find the cartridge is used as the sabot or the sabot is used on the cartridge.

Response 2: Update page 5 line 146-147. Further detail on the sabot cannot been provided due to industrial secrecy.

Point 3: On Fig. 4, the reviewer would like to know how to determine the residual velocities by the high-speed camera. Probably, it should be a magnitude. Additionally, the fragments will not perforate only one direction. The reviewer would like to know whether the camera can measure the three-dimensional components of velocity with the change in the angle after the perforation. Of course, as the authors mentioned, it is possible the fragments are shattered. Furthermore, the author mentioned “the high-speed camera recordings showed that the residual velocities of the shattered parts of the fragment are comparable with the residual velocities of the plugs and the energy release front travels with the same velocities of the shattered fragment and the plug” on Fig. 8. Detailed explanations how to measure the velocity should be given.

Response 3: Figure 5 (ex-Figure 4) in page 6 has been modified and updated. 
A discussion on the measurement of residual velocity has been inserted in page 3-4 line 116-130. A Figure (Fig. 3) has been added in page 4 as visual aid to the explanation.
The residual velocity in this context is considered the component of the velocity vector normal to the plate, i.e. parallel to the initial trajectory of the fragment. In Figure 5, the experimental residual velocities registered using the high-speed recordings are indicated by the square makers, while the dashed lines indicate the Recht-Ipson curve fits. 
With the experimental setup available, it is only possible to measure the pitch angle, while the yaw angle cannot be measured due to the 2D nature of the recordings. However, from the videos it is clearly visible that the yaw angle does not change significantly during the flight. This is object of further investigation using numerical modelling.

Point 4: In Fig. 4, the fitted curve cannot be drawn because CIV in Eq. (1) is unknown. The definition of CIV by Eq. (2) should be earlier

Response 4: The definition of CIV has been moved in page 6 line 183– 194;

Point 5: The explanations on Table 3, Fig. 7, 9, 10, 12 and 15 are insufficient. More information is required. What is the meaning on “X” and “V” in Table 3. The authors are just showing the figure and let the readers understand what is the meaning and phenomena

Response 5: 
Table 3, the meaning of X and V is defined in page 9 line 285-286; the data analysis is discussed in the following sections of the paper.
Figure 8 (ex-Figure 7), updated page 13 line 329-335;
Figure 10 (ex-Figure 9), update page 15 line 364-368;
Figure 11 (ex-Figure 10), caption updated page 16; update page 16 line 380-383;
Figure 13 (ex-Figure 12), caption updated page17; update page 16 line 397-400 and caption;
Figure 15, update page 21 line 467-474;

Point 6: In Fig. 5, the result of 3.5 mm MREs should be discussed by comparing the results of 5 mm MREs as well as 4.6 mm steel. Furthermore, the details for calculation the dashed line should be clear.

Response 6: Figure 6 (ex-Figure 5), caption updated; page 12 line 300-311: the dashed lines represent simply the plots of the kinetic energies for fragments having different masses, as also indicated in the legend of the Figure. Caption and picture have been updated.

Point 7: In Fig. 8, the figure is unclear to get sufficient information. Halation is happening in Fig. (e) and (f). Fig. (d) is so dark to identify where are the plug and fragment.

Response 7: Figure 9 (ex-Figure 8) has been modified, page 14.

Point 8: On Fig. 14, the reviewer would like to know how to determine the boundary between “Shear plugging” and “Transition” as well as “Transition” and “Adiabatic shearing”.

Response 8: Figure 15 (ex-Figure 14) update page 20 line 450-454; in Figure 14 the transition zone is indicated in grey (0.5<h/d<0.6). By reviewing publications on the formation and development of adiabatic shear banding it was concluded that it is difficult to define a clear boundary between shear plugging, transition and adiabatic shearing. The formation of ASB is an evolutionary process that starts from, approximately, 0.5 h/d. Furthermore, it is hard to define a specific transition value as results are strongly affected by the variability of microstructural morphology that is intrinsic in every commercial product. 
For instance, Xu et al. [3] note that microstructural inhomogeneity has a crucial role on the development of ABSs. The publication described the shear banding phenomenon as a process that progressively develops and progresses at a certain velocity of propagation. Two main types of localized shear bands were observed: deformed shear bands (dASB), which are characteristic for early stage deformations and shear localization, and transformed shear bands (tASB) that are created by further development of the dASB. The analysis of adiabatic shear bands in armour steel targets has been discussed in a dedicated section of the book by Couque [4]. In the case of intense interaction between projectile and plate, secondary shear bands were observed along the perforation channel. Yiadom et al. [5] observed that the evolution of the morphology of ASB is strain rate dependent. Higher impact momentums cause distinct shear bands, and, further augmenting the strain rates, cracks initiation and propagation, until final failure occurs. Recently, analysis on the development and the microstructural evolution of ASB in armor steel targets has been discussed by Jo et al. [6]. It was observed that during high velocity impacts the development of ASB is a progressive process and narrow tABS branch from wider bands.
Børvik et al. [7] [8] and Wen et al. [9] defined a range on h/d in which the transition happens. The considerations discussed in the paper under review are perfectly aligned with the observations of Børvik et al. [7] and Wen et al. [9], indicating a transition interval ranging between 0.5 and 0.6. A value of h/d 0.57 was statistically determined and indicated as threshold. However, as already stated, it is hard to define a specific transition value as results are strongly affected by the variability of microstructural morphology that is intrinsic in every commercial product.
The considerations discussed in the paper under review are supported by metallographic analyses.

•    Minor

Point 1: In line 134 on page #4, the reviewer cannot identify “calibre cartridge” is a proper technical word or not. The word “cartridge by caliber” can be found.

Response 9: Corrected and updated, page 5 line 151.

Point 2: In line 177 on page #5, the “red” should be replaced by “black”. In addition, 20 degree should be changed to 25 degree.

Response 10: Corrected and updated, page 7 line 206-208.

REFERENCES

1.     J. A. Zukas, High velocity impact dynamics, Wiley, 1990.
2.     K. R. Bratton, K. J. Hill, C. Woodruff, L. L. Campbell, C. B. Cagle, M. L. Pantoya, J. Magallanes, J. Abraham and C. Meakin, “High velocity impact testing of intermetallic projectiles,” Journal of dynamic behavior of materials, vol. 6, pp. 236 - 245, 2020.
3.     Y. B. Xu, W. L. Zhong, Y. J. Chen, L. T. Shen, Q. Liu, Y. L. Bai and M. A. Meyers, “Shear localization and recrystallization in dynamic deformation of 8090 Al-Li alloy,” Materials Science and Engineering A, vol. 299, pp. 287-295, 2001. 
4.     H. Couque, “Adiabatic shear bands in penetrators and targets,” in Adiabatic Shear Localization, Elsevier, 2012, pp. 247-266.
5.     S. B. Yiadom, A. K. Khan and N. Bassim, “Effect of microstructure on the nucleation and initiation of adiabatic shear bands (ABSs) during impact,” Materials Science & Engineering A, vol. 615, pp. 373-394, 2014. 
6.     M. C. Jo, S. Kim, D. W. Kim, H. K. Park, S. S. Hong, H. K. Kim, H. S. Kim, S. S. Sohn and S. Lee, “Understanding of adiabatic shear band evolution during high-strain-rate deformation in high-strength armor steel,” Journal of alloys and compounds, vol. 845, 2020. 
7.     T. Børvik, J. R. Leinum, J. K. Solberg, O. S. Hopperstad and M. Langseth, “Observations on shear plug formation in Weldox 460 E steel plates impacted by blunt-nosed projectiles,” International journal of impact engineering, vol. 25, pp. 553-572, 2001. 
8.     T. Borvik, O. S. Hopperstad, M. Langseth and K. A. Malo, “Effect of target thickness in blunt projectile penetration of Weldox 460 E steel plates,” International journal of impact engineering, vol. 28, pp. 413 - 464, 2003. 
9.     H. M. Wen and W. H. Sun, “Transition of plugging failure modes for ductile metal plates under impacr by flat-nosed projectiles,” Mechanics based design of structures and machines, vol. 38, pp. 86-104, 2010.

Round 2

Reviewer 2 Report

Thank you very much for the authors' great efforts.

I agree to publish this version in Materials.